# Mayaro Virus Non-Structural Protein 2 Circumvents the Induction of Interferon in Part by Depleting Host Transcription Initiation Factor IIE Subunit 2

**DOI:** 10.3390/cells10123510

**Published:** 2021-12-12

**Authors:** Ray Ishida, Jamie Cole, Joaquin Lopez-Orozco, Nawell Fayad, Alberto Felix-Lopez, Mohamed Elaish, Shu Yue Luo, Olivier Julien, Anil Kumar, Tom C. Hobman

**Affiliations:** 1Department of Medical Microbiology & Immunology, Faculty of Medicine & Dentistry, University of Alberta, Edmonton, AB T6G 2E1, Canada; rishida@ualberta.ca; 2Department of Cell Biology, Faculty of Medicine & Dentistry, University of Alberta, Edmonton, AB T6G 2H7, Canada; jamie.cole@ucalgary.ca (J.C.); lopezoro@ualberta.ca (J.L.-O.); nawell@ualberta.ca (N.F.); felixlop@ualberta.ca (A.F.-L.); elaish@ualberta.ca (M.E.); 3Department of Biochemistry, Faculty of Medicine & Dentistry, University of Alberta, Edmonton, AB T6G 2H7, Canada; syluo@ualberta.ca (S.Y.L.); ojulien@ualberta.ca (O.J.); 4Li Ka Shing Institute of Virology, University of Alberta, Edmonton, AB T6G 2E1, Canada

**Keywords:** alphavirus, MAYV, nsP2, interferon, TFIIE2, Rpb1

## Abstract

Mayaro virus (MAYV) is an emerging mosquito-transmitted virus that belongs to the genus *Alphavirus* within the family *Togaviridae*. Humans infected with MAYV often develop chronic and debilitating arthralgia and myalgia. The virus is primarily maintained via a sylvatic cycle, but it has the potential to adapt to urban settings, which could lead to large outbreaks. The interferon (IFN) system is a critical antiviral response that limits replication and pathogenesis of many different RNA viruses, including alphaviruses. Here, we investigated how MAYV infection affects the induction phase of the IFN response. Production of type I and III IFNs was efficiently suppressed during MAYV infection, and mapping revealed that expression of the viral non-structural protein 2 (nsP2) was sufficient for this process. Interactome analysis showed that nsP2 interacts with DNA-directed RNA polymerase II subunit A (Rpb1) and transcription initiation factor IIE subunit 2 (TFIIE2), which are host proteins required for RNA polymerase II-mediated transcription. Levels of these host proteins were reduced by nsP2 expression and during infection by MAYV and related alphaviruses, suggesting that nsP2-mediated inhibition of host cell transcription is an important aspect of how some alphaviruses block IFN induction. The findings from this study may prove useful in design of vaccines and antivirals, which are currently not available for protection against MAYV and infection by other alphaviruses.

## 1. Introduction

Mosquito-transmitted viruses within the genus *Alphavirus* infect hundreds of thousands of people and animals each year, causing enormous morbidity and mortality [1,2]. Mayaro virus (MAYV) is an emerging alphavirus that is currently circulating in South America and the Caribbean and is maintained primarily via the sylvatic cycle [3]. Acute MAYV infection manifests with flu-like symptoms, but a significant proportion of patients develop chronic arthritis that can persist for years [4]. MAYV belongs to the Semliki Forest virus complex, which comprises other arthritogenic alphaviruses [5] including the chikungunya virus (CHIKV), which because it can spread via an urban cycle, has caused multiple large outbreaks across the globe [6,7]. There are justified concerns regarding large MAYV outbreaks because the virus can be transmitted by *Aedes* sp. and *Anopheles* sp. of mosquitoes, which are prevalent in urban settings across a wide geographical range [8,9,10]. Imported cases of MAYV infection have already been reported in Europe and North America and are expected to increase in the future [11,12,13]. While therapeutics and vaccine candidates against MAYV are being investigated [14,15,16,17,18], currently there are no specific antiviral measures that could be used in the case of a MAYV outbreak. As such, further understanding of how this virus affects vital host cell pathways may facilitate development of effective antivirals.

The interferon (IFN) response is a critical first line of defense against viral infections [19,20,21]. During the induction phase, viral RNAs are detected by cytoplasmic RNA helicases such as RIG-I (retinoic acid-inducible gene) and MDA5 (melanoma differentiation-associated protein 5), leading to activation of MAVS (mitochondrial antiviral signaling protein) followed by recruitment of TBK1 (TANK-binding kinase 1)/IKKε (IκB kinase ε) or IKKα/IKKβ. These kinases then, respectively, activate the transcription factors IRF3 or NFκB, leading to induction of IFNs [22]. In the signaling phase of the IFN response, secreted IFNs bind to cognate receptors on the cell surface, which then activate JAK (Janus kinase)-STAT (signal transducer and activator of transcription factor) signaling and ultimately expression of interferon-stimulated genes (ISGs). ISG expression creates an anti-viral state, as evidenced by the fact that replication of alphaviruses is dramatically inhibited in cells pre-treated with IFNs [23,24,25]. The fact that many viruses have evolved strategies to inhibit both IFN induction and signaling further illustrates the importance of this antiviral response [26].

While alphavirus RNA can be sensed by both RIG-I and MDA-5 during infection [27], secretion of IFNs is almost completely abrogated during infection of fibroblasts by multiple alphaviruses [28,29,30]. Several studies have revealed mechanisms used by alphaviruses to suppress IFN signaling (reviewed in [31]). For example, non-structural protein 2 (nsP2) proteins of CHIKV and Semliki Forest virus (SFV) dampen the IFN response by inducing degradation of Rbp1, a component of the RNA polymerase II (pol II) complex [32,33,34,35,36]. In addition, CHIKV nsP2 stimulates nuclear export of STAT1, which in turn blocks IFN signaling [37]. However, very little is known regarding how alphaviruses block IFN induction.

In this study, we investigated how MAYV infection affects the IFN induction pathway. Similar to what has been reported for other alphaviruses [28,29,38], MAYV infection efficiently suppressed induction and secretion of IFN. Mapping studies indicated that nsP2 of MAYV blocks a step in the IFN induction pathway that is downstream of IRF3 phosphorylation. Interactome analysis revealed that nsP2 interacts with and causes depletion of the host protein TFIIE2, which is required for pol II-mediated transcription [39]. Loss of TFIIE2 was observed in cells infected with other alphaviruses within the Semliki Forest complex, suggesting that targeting this host protein is a common strategy used by these viruses to inhibit the IFN response.

## 2. Materials and Methods

### 2.1. Cell Culture and Virus Infection

Vero, A549, and HEK 293T cells from the American Type Culture Collection (Manassas, VA, USA) were cultured in Dulbecco’s modified Eagle’s medium (DMEM) (Gibco; Waltham, MA, USA) supplemented with 100 U/mL penicillin and streptomycin, 1 mM HEPES (Gibco), 2 mM glutamine (Gibco), and 10% heat-inactivated fetal bovine serum (FBS) at 37 °C in 5% CO_2_. C6/36 cells were kindly provided by Dr. Sonja Best, NIH Rocky Mountain laboratories (Hamilton, MT, USA) and was cultured in Minimal Essential Medium (MEM; Gibco) supplemented with 100 U/mL penicillin and streptomycin, 2 mM glutamine (Gibco), 10% heat-inactivated fetal bovine serum (FBS; Gibco), and 1× non-essential amino acids (Gibco) at 32  °C in 5% CO_2_. The Mayaro virus (MAYV) serotype D (strain 07-18066-99) was kindly gifted by Brandy Russell at Centre for Disease Control and Prevention (Fort Collins, CO, USA). Plasmid encoding Semliki Forest virus (strain SFV6.1) was a kind gift from Dr. Andres Merit at University of Tartu (Tartu, Estonia). Sindbis virus (SINV; Toto 1101) was kindly gifted by Dr. Charles Rice at Washington University School of Medicine (St. Louis, MO, USA). All MAYV, SFV, and SINV manipulation was performed according to level-2 containment procedures at University of Alberta. Virus stocks were generated in C6/36 cells and titrated using Vero cells. Sendai virus (Cantell Strain #10100774) was purchased from Charles River Laboratories (Wilmington, MA, USA).

### 2.2. Plasmids and Transfection

MAYV protein expression plasmids were generated from cDNA templates prepared by reverse transcription and polymerase chain reaction (RT-PCR) using RNA isolated from MAYV infected Vero cells. Viral gene-specific cDNAs including C-terminal 3×FLAG tag cassettes were cloned between NheI and FseI restriction sites in pcDNA 3.1(−) 3×FLAG plasmid [40]. Due to the auto-protease activity of capsid protein, which cleaved off the C-terminal 3×FLAG tag, a 3×FLAG sequence was added in frame to the N-terminal region of the protein. The primers used for the cloning are listed in Appendix A. The sequence integrity of all constructs was confirmed by Sanger sequencing.

For indirect immunofluorescence analysis, transfection of the appropriate expression plasmids into A549 cells was performed using TransIT-LT1 from Mirus Bio (Madison, WI, USA). For luciferase reporter assays in A549 or HEK 293T cells, plasmid transfection was performed using Lipofectamine 2000 from Invitrogen (Waltham, MA, USA).

### 2.3. Antibodies and Compounds

Rabbit anti-CHIKV-capsid and rabbit anti-SFV-capsid were kindly provided by Dr. Andres Merit, University of Tartu (Tartu, Estonia). Mouse anti-CHIKV-E2 hybridoma was a gift from Dr. Michael Diamond from Washington University School of Medicine (St. Louis, MO, USA). Commercially available antibodies were purchased from the following sources: mouse anti-β-actin (A3853) and mouse anti-FLAG (F3165) from Sigma-Aldrich (St. Louis, MO, USA); rabbit anti-IRF-3 (#11904) from Cell Signaling (Danvers, MA, USA); mouse anti-Rpb1 (GT9010) from GeneTex (Irvine, CA, USA); rabbit anti-GTF2E2 (ab187143) from Abcam (Cambridge, United Kingdom); poly(I:C) HMW (tlrl-pic) from InvivoGen (San Diego, CA, USA).

### 2.4. Immunoblotting

HEK293T cells (5.0 × 10^5^) or A549 cells (1.5 × 10^5^) seeded into 12-well plates were infected the next day with MAYV, SFV, or SINV, or transfected with protein expression plasmids. At experimental endpoints, cells were washed twice with phosphate-buffered saline (PBS) and lysed using 2× SDS sample buffer (62.5 mM Tris-HCl (pH 6.8), 50% (*v*/*v*) glycerol, 2% (*w*/*v*) SDS, 0.01% (*w*/*v*) Bromophenol blue) with 100 mM dithiothreitol (DTT). Cell lysates were incubated at 98 °C for 10 min to ensure denaturation of proteins. Samples were separated by molecular weight using SDS-PAGE and transferred to polyvinylidene difluoride (PVDF) membranes. The membranes were incubated in blocking buffer (PBS with 5% bovine serum albumin (BSA; Sigma Aldrich) and 0.05% Tween 20) for 30 min, after which they were incubated with primary antibodies diluted in blocking buffer for 60 min. Following three 10 min washes with PBS-0.05% Tween 20, the membranes were incubated with secondary antibodies, diluted in blocking buffer for 60 min, and protected from the light. The membranes were then subjected to three washes with PBS-0.05% Tween 20 and once with PBS. Protein bands were imaged using an Odyssey Infrared Imaging system. Quantification of proteins was performed using Odyssey Image Studio Lite Software Version 5.2.

### 2.5. Immunoprecipitation

HEK293T cells (3.6 × 10^5^) in T25 were transfected with 6 µg of indicated plasmids using Lipofectamine 2000 (Life Technologies; Carlsbad, CA, USA), according to the manufacturer’s protocol. Twenty-four hours later, cells were harvested and resuspended in ice-cold IP buffer (150 mM NaCl, 50 mM Tris (pH 7.5), 1% Triton X-100, 1 mM NaF, 1 mM DTT, and protease inhibitor cocktail (Roche; Basel, Switzerland)). After 30 min, lysates were clarified by centrifugation at 16,000× *g* for 15 min and then incubated with 10 µL of anti-FLAG M2 magnetic beads (Millipore; Burlington, MA, USA) at 4 °C for 2 h. After washes with IP buffer, bound proteins were analyzed by mass spectrometry (LC–MS/MS) or SDS sample buffer was added to the beads, which were then incubated 98 °C for 10 min to denature proteins, followed by SDS-PAGE.

### 2.6. Mass Spectrometry and Mass Spectrometry Interaction Statistics (MiST)

Trypsin digestions of proteins immunoprecipitated using anti-FLAG magnetic resins were performed on a KingFisher Duo Prime (Thermo Scientific; Waltham, MA, USA). Briefly, the samples were reduced (10 mM DTT in 50 mM ammonium bicarbonate) and alkylated (50 mM iodoacetamide in 50 mM ammonium bicarbonate) before digestion for 5 h at 37 °C using sequencing grade modified trypsin (Promega; Madison, WI, USA). The tryptic peptides were separated using an Easy-nLC 1000 liquid chromatograph (Thermo Scientific) and an EASY-Spray capillary column (ES900, Thermo Scientific). Mass spectrometry was performed on a Q Exactive Orbitrap mass spectrometer (Thermo Scientific). The mass spectrometer was operated in data-dependent acquisition mode with a resolution of 35,000 and *m*/*z* range of 300–1700. The 12 most intense multiply charged ions were sequentially fragmented by HCD dissociation, and spectra of their fragments were recorded in the orbitrap at a resolution of 17,500, followed by a 30 s dynamic exclusion. Data were processed using Proteome Discoverer 1.4 (Thermo Scientific), and databases were searched using SEQUEST (Thermo Scientific). Search parameters included a strict false discovery rate (FDR) of 0.01, a relaxed FDR of 0.05, a precursor mass tolerance of 10 ppm, and a fragment mass tolerance of 0.01 Da. Peptides were searched with carbamidomethyl cysteine as a static modification and oxidized methionine, and deamidated glutamine and asparagine as dynamic modifications.

The results from mass spectrometry were analyzed for protein–protein interaction scoring with MiST (mass spectrometry interaction statistics) pipeline to generate the interactome map of each viral protein. Protein names, functions, and putative cellular localizations were extracted from the Uniprot database [41].

### 2.7. Biological Process Enrichment Analysis

Biological process enrichment analysis and generation of resulting heatmaps were performed using Metascape (http://metascape.org, accessed on 30 June 2021]) [42]. Host proteins found to interact with MAYV nsP2 with a MiST score of ≥0.80 were analyzed. The ontology sources GO biological processes, GO cellular components, KEGG structural complexes, and CORUM were used for the enrichment analysis. Metascape analyses generated accumulative hypergeometric *p*-values, q-values using the Benjamini–Hochberg procedure, and enrichment factors for filtering of the gene list. The following thresholds were used: *p*-value < 0.01, enrichment factor > 1.5, and a minimum overlap set at a count of 3. Statistically significant enriched terms were then hierarchically clustered using Kappa similarity scoring; a Kappa score > 0.3 was used to define clusters. Within each cluster, the most significantly enriched term was chosen and displayed in a heatmap.

### 2.8. Confocal Immunofluorescence (IF) Microscopy

A549 cells on coverslips were fixed with 4% electron microscopy-grade paraformaldehyde (Electron Microscopy Sciences; Hatfield, PA, USA) in PBS for 10 min at room temperature and permeabilized with 0.5% Triton X-100 in PBS for 10 min. Coverslips were then washed three times with PBS, after which they were incubated in blocking buffer for 30 min at room temperature. Coverslips were then incubated with primary antibodies diluted in blocking buffer for 45 min at room temperature. Coverslips were washed three times with PBS, and then they were incubated with secondary antibodies (1:1000; Invitrogen) and DAPI (4′,6-diamidino-2-phenylindole; 1 μg/mL; Sigma Aldrich) in blocking buffer for 45 min at room temperature. Samples were then washed three times with PBS and once with de-ionized water; then, they were mounted onto microscope slides with Prolong Gold anti-fade mounting reagent (Life Technologies). Samples were imaged using an Olympus IX-81 spinning-disk confocal microscope equipped with a 60× PlanApo N oil objective. Images were analyzed using Volocity 6.2.1 software (PerkinElmer; Waltham, MA, USA).

### 2.9. Quantitative Real Time PCR (qRT-PCR)

Intracellular RNA was extracted from cells using the RNA NucleoSpin Kit (Macherey Nagel; Duren, Germany) and reverse transcribed into cDNAs by incubating the isolated RNA with random primers (Invitrogen) and Improm-II reverse transcriptase (Promega) at 42 °C for 1.5 h. The cDNAs were diluted 1:5 with pure water, and 5% of the volume was mixed with the appropriate primers (Integrated DNA Technologies; Coralville, IA, USA) and the PerfecTa SYBR green SuperMix with Low ROX (Quanta Biosciences; Gaithersburg, MD, USA) and amplified for 40 cycles (30 s at 94 °C, 40 s at 55 °C, and 20 s at 68 °C) in a Biorad CFX96 qRT-PCR machine. The target genes and primer sequences used are listed in Appendix A. The CT values were normalized with *Actb* mRNA as the internal control or by cell count (ΔCT). The ΔCT values were further normalized to ΔCT values of control samples (ΔΔCT). The relative mRNA levels were calculated using the formulas 2^(−ΔΔCT)^.

### 2.10. Luciferase Reporter Assay

HEK293T cells (4.5 × 10^5^) seeded in 12-well plates were transfected with the following promoter reporter (*Firefly* luciferase) constructs IFN-β: p125-luc and IRF3: p55-CIB-Luc (provided by T. Taniguchi, University of Tokyo, Japan), NFκB: or pNF-kB-Luc (Stratagene), together with the *Renilla* luciferase: pRL-TK (Promega) as a transfection control. At experimental endpoints, cells were washed once with PBS and then lysed with 250 μL Luciferase Lysis buffer (0.1% (*v*/*v*) Triton X-100, 25 mM glycylglycine (pH 7.8), 15 mM MgSO_4_, 4 mM EGTA, and 1 mM DTT), after which the samples were analyzed immediately or stored at −80 °C until further use. For luciferase assays, 50 μL of samples were aliquoted into white 96-well microplates (Greiner bio-one) in duplicates for both *Firefly* and *Renilla* luciferase activity measurements. The *Firefly* luciferase substrate D-luciferin (Gold Biotechnology; St. Louis, MO, USA) was prepared at a final concentration of 70 µM in luciferase assay buffer (25 mM glycylglycine (pH 7.8), 15 mM K_2_PO_4_ (pH 7.8), 15 mM MgSO_4_, 4 mM EGTA, 1 mM DTT, and 2 mM ATP), and 100 μL was added to each well; then, they were incubated for 5 min before the luciferase activity was measured using a Synergy HTX plate reader (Biotek; Winooski, VT, USA). For *Renilla* luciferase, the substrate coelenterazine (Gold Biotechnology USA) was diluted to a final concentration of 1.4 µM in luciferase assay buffer (25 mM glycylglycine (pH 7.8), 15 mM K_2_PO_4_ (pH 7.8), 15 mM MgSO_4_, and 4 mM EGTA). One-hundred microliters were added to each well, and luciferase activity was measured immediately using a Synergy HTX plate reader (Biotek).

### 2.11. Enzyme-Linked Immunosorbent Assay (ELISA)

Levels of human IFN-β in the cell culture supernatant were measured using Quantikine^®^ Human IFN-β Immunoassay kit (R&D Systems, Inc.; Minneapolis, MN, USA), as per the manufacturer’s instructions. The total fluorescence was measured using a Synergy HTX plate reader (Biotek).

### 2.12. Statistical Analyses

All statistical analyses were performed using Microsoft Excel. Two-tailed paired Student’s *t*-test was performed for pair-wise statistical comparison. One-way ANOVA followed by two-tailed paired Student’s *t*-test was used for comparison of multiple samples. The individual data points from each independent experiment and mean ± standard error of the mean are shown in the bar and line graphs.

## 3. Results

### 3.1. MAYV Suppressed Production of Type I and III IFNs Downstream of IRF3 Nuclear Translocation

To determine if/how IFN induction is affected during MAYV infection, we quantitated type I IFN (*Ifnb*) and viral genomic RNAs by qRT-PCR at 4, 8, 12, and 24 h after A549 cells were infected with MAYV or Sendai virus (SeV). Despite robust virus replication, induction of *Ifnb* mRNA in MAYV-infected cells was delayed and dramatically suppressed compared to cells infected with SeV (Figure 1A,B), a potent inducer of IFN production [43]. However, treatment of cells with type I IFN prior to infection significantly inhibited MAYV in a dose-dependent manner (Figure 1C), indicating that the virus is sensitive to IFN.

Next, we assessed whether the relatively low levels of *Ifnb* mRNA in MAYV-infected cells were due to active suppression of IFN induction pathways. MAYV-infected cells were transfected with poly(I:C), a dsRNA analog that induces IFN following detection by RIG-I-like receptors [44]. Transcripts encoding type I (*Ifnb*) and type III (*Ifnl2*) IFNs were then quantified by qRT-PCR. Poly(I:C) transfection robustly induced *Ifnb* and *Ifnl2*, but levels of these transcripts were ≈50-fold lower in MAYV-infected cells compared to mock-infected cells (Figure 1D,E). MAYV infection alone resulted in a relatively modest increase (≈100-fold) in *Ifnb* and *Ifnl2*, which were further increased ≈10-fold by poly(I:C) challenge. In comparison, SeV infection resulted in ≈10,000-fold increased levels of *Ifnb*, but this effect was dampened in cells that were first infected with MAYV (Figure 1F). IFN-β was not detected in the media of MAYV-infected cells, regardless of whether they were challenged with SeV or not (Figure 1G). The observation that levels of SeV genomic RNA were higher in MAYV-infected cells suggests that impaired IFN induction during MAYV infection was not due to poor replication of SeV in those cells (Figure 1H).

Next, we examined whether nuclear translocation of the antiviral transcription factor IRF3, which is required for production of IFN-β mRNA, was affected by MAYV infection. In contrast to SeV infection, which induced translocation of IRF3 into nuclei, IRF3 remained in the cytoplasm of MAYV-infected cells (Figure 1I). However, when MAYV-infected cells were subsequently infected with SeV, IRF3 localized to the nucleus (Figure 1I). This indicates that the IFN induction pathway is intact in MAYV infected cells up to and including the IRF3 nuclear translocation step. Taken together, these data are compatible with a scenario where MAYV efficiently avoids detection by RIG-I-like receptors, and/or partially blocks a step upstream of IRF3 nuclear transport to prevent IFN production. Because SeV-induced IRF3 nuclear translocation is unaffected in MAYV-infected cells (Figure 1I), but production of IFN mRNA and secretion of the cytokines are blocked (Figure 1D–G), the virus likely targets additional steps in the IFN induction pathway downstream of IRF3 nuclear translocation.

### 3.2. MAYV nsP2 Abrogated Interferon Induction Downstream of IRF3 Activation

To identify the viral protein(s) responsible for suppression of the IFN induction pathway during MAYV infection, we generated expression plasmids encoding individual epitope-tagged MAYV proteins. With the exception of the capsid protein, which was tagged on the N-terminus, all MAYV proteins were constructed with a 3×FLAG epitope at their C-termini. It was not feasible to tag the C-terminus of MAYV capsid because, like other alphavirus capsid proteins, it has auto-protease activity that cleaves the hydrophobic signal peptide at its C-terminus [45]. Expression of the tagged viral proteins in transfected HEK293T cells was authenticated by immunoblotting (Figure 2A).

Next, the effect of MAYV protein expression on IFN induction in response to SeV infection was evaluated using an IFN-β-promoter-based luciferase reporter assay. The only MAYV protein whose expression significantly blocked IFN-β promoter activity was nsP2 (Figure 2B). Despite its relatively low expression compared to the other FLAG-tagged MAYV proteins (Figure 1A), nsP2 suppressed IFN-β promoter activity almost as efficiently as the NS1 protein of influenza A virus, a known suppressor of the IFN induction [46] and the positive control for these experiments. Because activation of IRF3 and NFκB is critical for induction of *Ifnb* [21], we assessed whether nsP2 abrogates the activities of these two transcription factors using IRF3 and NFκB promoter-based luciferase reporters. Compared to the vector control and MAYV capsid protein, nsP2 reduced IRF3- and NFκB-dependent luciferase expression by as much as 10-fold (Figure 2C,D). To determine the step in the IFN induction pathway targeted by nsP2, we assessed how overexpressing individual components in the pathway (RIG-I (2xCARD), TBK1, IKKɛ, IRF3, and constitutively active IRF3 (IRF3-5D)) affected nsP2-dependent inhibition. None of these components rescued the nsP2 blockade of the IFN induction pathway (Figure 2E), suggesting that this viral protein acts downstream of the IRF3 phosphorylation step. Interestingly, IRF3 nuclear translocation in response to SeV infection was not affected by nsP2 expression (Figure 2F,G). These data are consistent with those shown in Figure 1 in which MAYV infection was able to block IFN induction downstream of IRF3 nuclear translocation.

Alphavirus nsP2 proteins contain a helicase domain at the N-terminus with RNA-dependent NTPase activity [47,48], a papain-like cysteine protease domain [49], and a C-terminal methyltransferase domain [50] (Figure 2H). To determine which domain(s) of nsP2 were important for blocking IFN induction, we generated expression constructs encoding nsP2 lacking NTPase (nsP2^K197N^) [48], protease activity (nsP2^C478A^) [49], helicase domain (nsP2^Δhelicase^), or protease domain (nsP2^Δprotease^) (Figure 2H). Both nsP2^K197N^ and nsP2^C478A^ reduced IFN induction in response to SeV infection, similar to wildtype nsP2 and influenza A virus NS1 protein (Figure 2I), indicating that NTPase and protease activities are not required to block IFN induction. However, constructs lacking the protease (nsP2^Δprotease^) or helicase (nsP2^Δhelicase^) domains were unable to block IFN induction in response to SeV infection (Figure 2I).

### 3.3. The Effect of nsP2 on Interferon Induction Was Found to Be Partially Mediated by Transcriptional Shutoff

Previous studies have shown that other alphavirus nsP2 proteins translocate to the nucleus and abrogate host transcription by depleting Rbp1, a component of the RNA polymerase II complex [33,51]. To determine if these processes were important for the ability of MAYV nsP2 to antagonize IFN induction, we used site-directed mutagenesis to generate nuclear localization signal (NLS)-deficient (nsP2^RK653AA^) [36] and transcriptional shutoff-deficient (nsP2^P722S^) [51] mutants of nsP2 (Figure 3A). Confocal microscopy analyses confirmed that nsP2^RK653AA^ was not able to translocate into the nucleus, whereas wildtype and nsP2^P722S^ were detected in nuclei and the cytoplasm (Figure 3B). However, while both nsP2^RK653AA^ and nsP2^P722S^ significantly blocked IFN induction, nsP2^P722S^ was around fourfold less effective than wildtype nsP2 (Figure 3C). Furthermore, in cells over-expressing a constitutively active form of IRF3 (IRF3-5D) and nsP2^RK653AA^ or nsP2^P722S^, levels of IFN transcripts were higher than in cells expressing wild type nsP2 (Figure 3E). Unlike, wildtype nsP2 and nsP2^RK653AA^, however, nsP2^P722S^ did not reduce IRF3-5D expression in co-transfected cells (Figure 3F,G). While these data suggested that nsP2 suppresses IFN induction in part by mediating host cell transcriptional shutoff, it also appears to function through a second mechanism that is independent from its nuclear function or ability to abrogate transcription.

### 3.4. nsP2 Interacted with and Downregulated the Levels of Host Transcription Mediators Rpb1 and TFIIE2

To further investigate how MAYV affects the host antiviral response, we used co-immunoprecipitation (co-IP) followed by tandem mass spectrometry (LC–MS/MS) to identify host cell proteins that interact with nsP2. Putative nsP2-binding host proteins were subjected to analyses using Molecular Interaction Search Tool (MiST) (Appendix A), and key cellular processes associated with nsP2 interactors were assessed through Gene Ontology (GO) enrichment analysis. Surprisingly, we did not detect any protein directly involved in the IFN induction pathway interacting with nsP2 of MAYV. One of the key enriched GO terms was “RNA polymerase II holoenzyme complex” (Figure 4A, Appendix A), which consists of proteins involved in transcription by RNA polymerase II. Two members of this complex were identified in the nsP2 co-IP: specifically, Rpb1 (also known as POLR2A), a component of RNA polymerase II that was previously shown to interact with other alphavirus nsP2 proteins [33,51], and TFIIE2, an integral factor that functions in transcription initiation [39]. Until this study, TFIIE2 was not known to interact with alphavirus nsP2. Interestingly, the transcriptional shutoff mutant nsP2^P722S^ formed a stable interaction with Rpb1 but not TFIIE2 (Figure 4B,C). Confocal microscopy analyses revealed that in cells transfected with nsP2, expression levels of Rpb1 and TFIIE2 were significantly reduced (Figure 4D–G). The NLS mutant nsP2^RK653AA^ reduced TFIIE2 but not Rpb1 levels, whereas the transcriptional shutoff mutant nsP2^P722S^ did not affect levels of either protein (Figure 4D–G). Accordingly, nuclear localization of nsP2 appears to be important for depleting Rpb1 but not for suppressing TFIIE2 expression.

### 3.5. Some but Not All Alphaviruses Induced Loss of TFIIE2

Quantitative confocal analyses confirmed that MAYV infection resulted in significant loss of Rpb1 and TFIIE2 proteins at 24 hpi (Figure 5A–D). To elucidate how this occurs, we treated MAYV-infected cells with inhibitors of proteasomal- (epoxomicin) and lysosomal- (bafilomycin A1) dependent degradation followed by immunoblot analyses. Loss of Rpb1 protein during MAYV infection was significantly inhibited by epoxomicin (Figure 5E,F), which is consistent with previous findings that old world alphavirus nsP2-dependent depletion of Rpb1 involves the proteasome [51]. Conversely, neither epoxomicin nor bafilomycin blocked degradation of TFIIE2 protein during infection with MAYV (Figure 5E–G). Next, we examined the levels of Rpb1 and TFIIE2 proteins in SFV- and Sindbis virus (SINV)-infected cells. Similar to what was observed during MAYV infection, Rpb1 and TFIIE2 protein levels were both reduced at 24 hpi in SFV-infected cells (Figure 5H–J). However, in cells infected with SINV, Rpb1 but not TFIIE2 levels were lower (Figure 5H–J).

Finally, as a first step toward understanding how MAYV infection affects host cell transcription, we assessed global levels of RNA as well as transcripts of housekeeping genes following infection. As expected, total cellular RNA was greatly reduced at 48 hpi (Figure 6A), while housekeeping gene transcripts were lower at 24 hpi (Figure 6B–D). One possibility to explain the lack of substantial decrease in total RNA level at 24 hpi may be due to increasing levels of viral RNA during this time period. Indeed, we observed a significant increase in viral RNA from 8 to 24 hpi (Figure 6E). Together, these results suggested that MAYV infection abrogates IFN induction in part by blocking global transcription through nsP2-mediated depletion of transcription factors TFIIE and Rpb1.

## 4. Discussion

The IFN response is critical for controlling alphavirus infections, including the emerging mosquito-transmitted pathogen MAYV [52,53,54,55]. While considerable efforts have been directed at understanding how alphaviruses affect the downstream signaling phase of the IFN response [24,33,37], relatively little is known regarding how production of IFN is dampened during infection. Our analyses revealed that compared to SeV infection, induction of type I IFN (*ifnb*) transcripts in MAYV-infected cells was dramatically suppressed and significantly delayed. These findings are consistent with earlier studies on other alphaviruses such as SINV, CHIKV, Ross River virus, Venezuelan Equine Encephalitis virus, and Eastern Equine Encephalitis virus [28,38]. Similarly, production of type III IFN transcripts was inhibited, and secretion of IFN-β was completely abrogated by MAYV infection. These suggest that MAYV suppresses induction of transcripts and may further impede translation and/or secretion of IFNs. Impaired IFN secretion from alphavirus-infected epithelial cells has been reported [28,29,30], but interestingly, this does not appear to be the case during infection of monocytes with Venezuelan Equine Encephalitis virus [29]. Accordingly, it will be of interest to study the IFN response during MAYV infection of key target cells, such as synovial fibroblasts, monocytes, and macrophages in future.

Consistent with earlier studies showing that alphavirus nsP2 proteins are involved in antagonizing the IFN response [24,32,33,36,37,56], we observed that nsP2 of MAYV is highly effective at blocking IFN induction. Of note, Bae et al. showed that CHIKV envelope protein also inhibited IFN induction [56], but in our assays, nsP2 was the only MAYV protein that blocked this process. This may be due to inherent differences between MAYV and CHIKV envelope proteins themselves and/or the fact that our study employed constructs comprised of original viral sequences rather than codon optimized plasmids as used by Bae et al. [56]. Finally, SINV transframe protein, which is produced when a frameshift occurs during translation of 6K protein, has been reported to antagonize the IFN response [57]. However, whether the analogous protein of MAYV or other alphaviruses functions in a similar manner remains to be determined.

Our analysis showed that MAYV nsP2 suppresses IFN induction downstream of IRF3 phosphorylation and nuclear translocation, which may indicate that transcription and/or translation of IFN mRNAs is affected. Transcriptional shutoff by other alphavirus nsP2 proteins is important for circumventing the IFN response, particularly the signaling arm of this pathway [32,33,34,35]. The fact that the transcriptional shutoff defective mutant of MAYV nsP2 (nsP2^P722S^) was less effective at blocking IFN induction is consistent with these previous studies. While a few studies have indicated that alphavirus nsP2 is responsible for hindering host translation during infection [34,58], others suggest that the translational shutoff observed during infection is due to sequestration of translational machinery by viral RNA [59,60]. Nevertheless, since we observed that MAYV completely blocked the translation and or/secretion of IFNs, future studies are required to determine whether nsP2 suppresses translation of IFNs.

Alphavirus nsP2 proteins have been shown to abrogate host transcription in part by depleting Rpb1, a component of the RNA polymerase II holoenzyme [33,51]. Interactome analyses showed that MAYV nsP2 binds Rbp1 and TFIIE2, host proteins that are important for initiation of RNA polymerase II-dependent transcription [39]. Loss of TFIIE2 was observed in MAYV-infected cells as well as cells expressing nsP2 alone. Interestingly, while cells infected with SFV also had lower levels of this protein, SINV infection had no such effect on TFIIE2. Reducing levels of TFIIE2 would be expected to reduce cellular transcription during MAYV infection, but it is unclear as to how this occurs, as neither inhibition of proteasome- nor lysosome-dependent degradation blocked virus-induced turnover of this host protein. Conversely, we observed that nsP2-dependent loss of Rpb1 is proteasome-dependent, similar to what has been reported for SFV and SINV infection [33,51]. If TFIIE2 degradation is important for blocking IFN induction during MAYV, it is unlikely that the protease activity of nsP2 is required for this process since the protease-dead nsP2^C478A^ mutant suppressed IFN induction just as well as wildtype nsP2. Moreover, TFIIE2 lacks the consensus cleavage sequence that is normally targeted by alphavirus nsP2 [61]. Thus, elucidating the mechanism by which nsP2 depletes TFIIE2 requires further investigations.

The observation that MAYV nsP2^P722S^ still inhibited IFN production, albeit not as well as wild type nsP2, may indicate that other mechanisms are at play. Interestingly, nsP2^P722S^ was able to form a stable complex with Rpb1, but not TFIIE2. This could mean that nsP2-TFIIE2 interaction is important for blocking host transcription or IFN induction by other means. Although expression of MAYV nsP2 resulted in depletion of both Rpb1 and TFIIE2, nuclear translocation of nsP2 was only required to reduce levels of Rpb1 but not TFIIE2. Prominent loss of TFIIE2 protein was also observed during SFV infection but not in cells infected with SINV. Additional studies are required to determine the impact of each Rpb1 and TFIIE2 depletion on host transcription and IFN induction.

In summary, the present study has revealed novel mechanisms by which MAYV subverts the innate immune response. Given that this emerging alphavirus has the potential to cause large multicontinental outbreaks similar to CHIKV [8,10,62], it is critical to understand how it manipulates host pathways during infection. Indeed, there is considerable interest in developing therapeutic and prophylactic therapies against MAYV [14,15,16,17,18]. The findings from the present study may be of use for development and application of therapeutics against MAYV and other alphaviruses.

## Figures and Tables

**Figure 1 cells-10-03510-f001:**
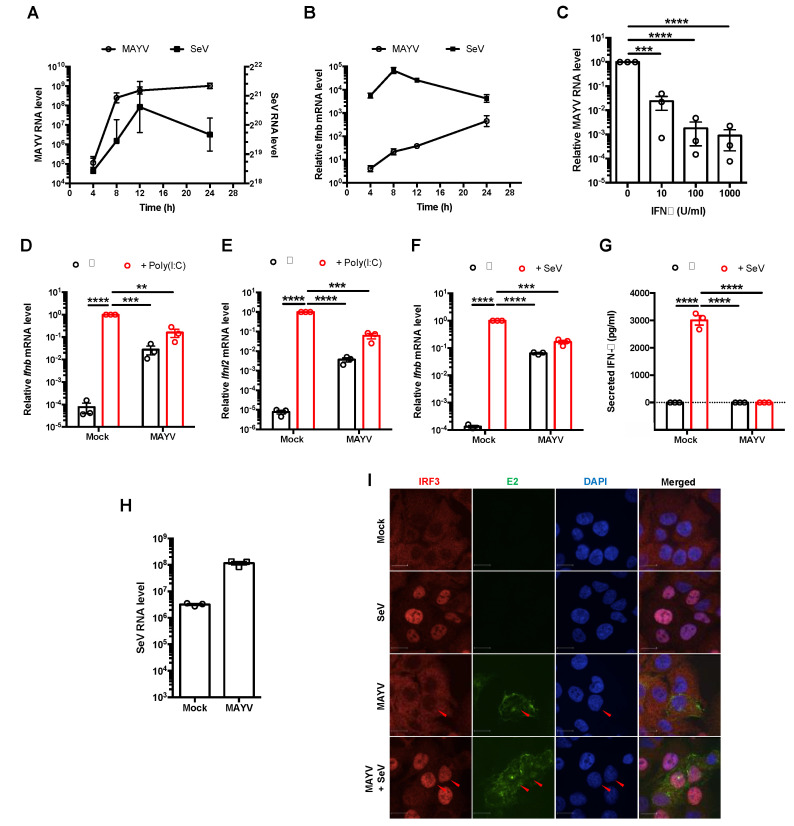
MAYV suppressed production of type I and III interferons. (**A**,**B**) A549 cells were infected with either MAYV (MOI = 3) or SeV (50 HAU/mL). Total RNA was collected at 4 h intervals for up to 24 h post-infection (hpi). Viral RNA (**A**) and *Ifnb* mRNA (**B**) were quantitated by qRT-PCR and expressed as fold mock infected cells normalized to *Actb* mRNA levels. Data shown are the mean ± SEM from three independent experiments. (**C**) A549 cells were treated with IFN-α (0, 10, 100, and 100 U/mL) for 6 h and infected with MAYV (MOI = 1). At 24 hpi, relative levels of MAYV RNA in total cellular RNA samples were quantitated by qRT-PCR (normalized to *Actb*) and expressed as folds of replication to cells not treated with IFN. Data represent the mean ± SEM from three independent experiments and were analyzed by Student’s *t*-test. *** *p* < 0.001, **** *p* < 0.0001. (**D**,**E**) A549 cells were infected with MAYV (MOI = 3) for 24 h, then treated with 2 µg/mL of poly(I:C). After 16 h, total RNA was collected, and *Ifnb* (**D**) and *Ifnl2* (**E**) mRNA were measured by qRT-PCR and normalized to *Actb* mRNA level and expressed as folds to mock infected cells. Data represent the mean ± SEM from three independent experiments and were analyzed by one-way ANOVA and Student’s *t*-test. ** *p* < 0.01, *** *p* < 0.001, **** *p* < 0.0001. (**F**,**G**) A549 cells were infected with MAYV (MOI = 3) for 24 h, then challenged with (SeV) (50 HAU/mL) for 16 h. *Ifnb* transcripts in cells and IFN-β in cell culture supernatants was measured by qRT-PCR (**F**) and enzyme-linked immunosorbent assay (**G**), respectively. Data represent the mean ± SEM from three independent experiments and were analyzed by one-way ANOVA and Student’s *t*-test. *** *p* < 0.001, **** *p* < 0.0001. (**H**) A549 cells were infected with MAYV (MOI = 3) for 24 h, then infected with SeV (50 HAU/mL) for 16 h. Total RNA was collected, and SeV viral RNA level was measured by real-time qRT-PCR and normalized to *Actb* mRNA level and expressed as folds of mock infected cells. Data are represented as mean ± SEM from three independent experiments. (**I**) A549 cells were infected with MAYV (MOI = 1) for 24 h and then infected with 50 HAU/mL of SeV for 8 h. The subcellular localization of IRF3 in MAYV-infected cells was visualized with a confocal microscope using 60× oil objective after staining with antibodies against IRF3 and MAYV E2 protein. Scale bar = 12 µm.

**Figure 2 cells-10-03510-f002:**
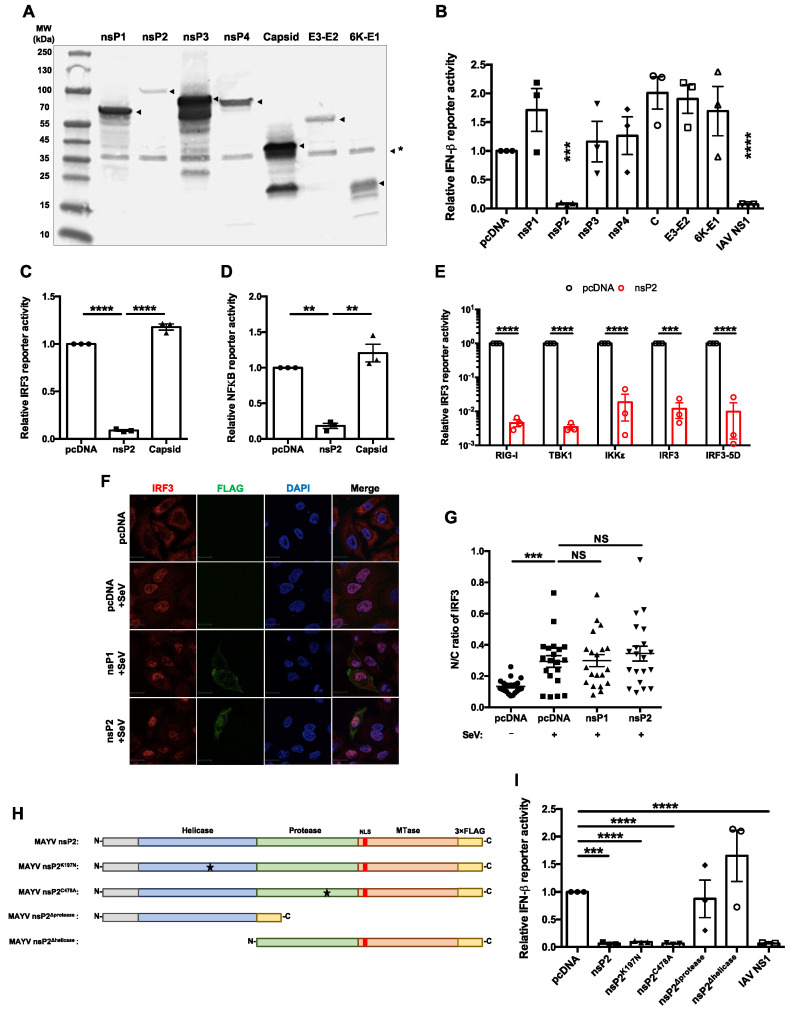
MAYV nsP2 abrogated interferon induction downstream of IRF3 activation. (**A**) HEK293T cells were transfected with pcDNA 3.1 (−) plasmids encoding the indicated 3×FLAG-tagged MAYV proteins. After 24 h, cell lysates were subjected to immunoblotting with anti-FLAG antibody. The positions of the epitope tagged viral proteins are indicated with arrowheads. A non-specific protein recognized by the anti-FLAG or secondary antibody is indicated by *. (**B**) HEK293T cells were co-transfected with plasmids encoding the indicated MAYV proteins, empty vector (pcDNA), or Influenza A virus NS1, as well as a plasmid encoding IFN-β *Firefly* luciferase and constitutively expressed control *Renilla* luciferase. After 24 h, cells were infected with 50 HAU/mL of SeV, and then *Firefly* and *Renilla* luciferase activities were measured 16 h later. Data represent the mean ± SEM from three independent experiments and were analyzed by Student’s *t*-test. *** *p* < 0.001, **** *p* < 0.0001. (**C**,**D**) HEK293T cells were co-transfected with plasmids encoding nsP2 or capsid proteins and *Firefly* luciferase under the control of IRF3- (**C**) or NFκB- (**D**) responsive promotors as well as a plasmid encoding constitutively expressed *Renilla* luciferase. After 24 h, cells were challenged with 50 HAU/mL of SeV for 16 h, after which *Firefly* and *Renilla* luciferase activities were measured. Data represent the mean ± SEM from three independent experiments and were analyzed by one-way ANOVA and Student’s *t*-test. ** *p* < 0.01, **** *p* < 0.0001. (**E**) HEK293T cells were co-transfected with plasmids encoding nsP2, and RIG-I (2xCARD), IKKε, TBK1, IRF3, IRF3-5D, or empty vector, IRF3-promotor *Firefly* luciferase and constitutively expressed *Renilla* luciferase. Samples were harvested at 24 h post-transfection, after which *Firefly* and *Renilla* luciferase activities were measured. Data represent the mean ± SEM from three independent experiments and were analyzed by Student’s *t*-test. *** *p* < 0.001, **** *p* < 0.0001. (**F**,**G**) A549 cells were transfected with indicated 3×FLAG-tagged protein-encoding plasmids. After 24 h, cells were challenged with SeV infection (50 HAU/mL) for 8 h, then fixed and stained with antibodies against FLAG and IRF3 and visualized with a confocal microscope using 60× oil objective. The cytoplasmic and nuclear IRF3 signals were quantified using Volocity software. Scale bar = 12 µm. Data represent the mean ± SEM from three independent experiments (*n* = 20) and were analyzed by one-way ANOVA and Student’s *t*-test. *** *p* < 0.001, NS = not significant. (**H**) Schematic of 3 × FLAG-tagged wild type nsP2, NTPase mutant nsP2^K197N^, protease-dead nsP2^C478A^, helicase only nsP2^Δprotease^, and protease only nsP2^Δhelicase^. (**I**) HEK293T cells were transfected with the indicated viral nsP2 constructs, IFN-β *Firefly* luciferase reporter, and a control *Renilla* reporter. After 24 h, cells were infected with 50 HAU/mL of SeV for 16 h, after which relative *Firefly* and *Renilla* luciferase activities were measured. Data represent the mean ± SEM from three independent experiments and were analyzed by one-way ANOVA and Student’s *t*-test. *** *p* < 0.001, **** *p* < 0.0001.

**Figure 3 cells-10-03510-f003:**
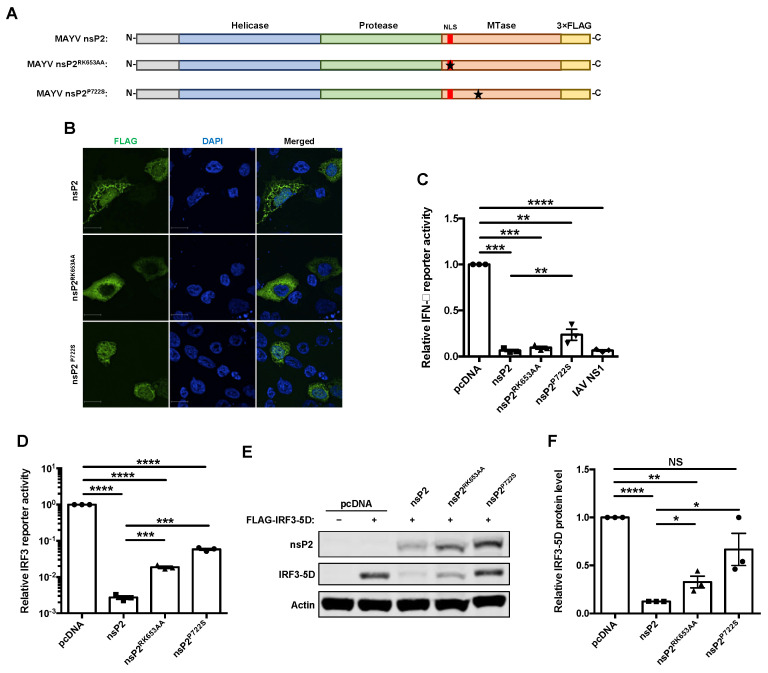
Inhibition of interferon induction by nsP2 was found to be partially mediated by transcriptional shutoff. (**A**) Schematic of 3×FLAG-tagged wildtype nsP2, NLS-deficient nsP2^RK653AA^, and host transcription shutoff deficient nsP2^P722S^. (**B**) A549 cells were transfected with indicated MAYV nsP2 constructs. After 24 h, cells were fixed and stained using α-FLAG antibody and imaged visualized with a confocal microscope using 60× oil objective. Nuclei are stained with DAPI. Scale bar = 12 µm. (**C**) HEK293T cells were transfected with the indicated viral nsP2 constructs, IFN-β *Firefly* luciferase reporter, and a control *Renilla* reporter. After 24 h, cells were infected with 50 HAU/mL of SeV for 16 h, after which relative *Firefly* and *Renilla* luciferase activities were measured. Data represent the mean ± SEM from three independent experiments and were analyzed by one-way ANOVA and Student’s *t*-test. ** *p* < 0.01, *** *p* < 0.001, **** *p* < 0.0001. ((**D**–**F**) HEK293T cells were transfected with indicated viral nsP2 constructs, FLAG-IRF3-5D, IRF3-promotor *Firefly* luciferase reporter, and a control *Renilla* reporter. After 24 h, *Firefly* and *Renilla* luciferase activities were measured (**D**). Cell whole lysate were also analyzed by immunoblotting using antibodies against FLAG and actin (**E**,**F**). Data represent the mean ± SEM from three independent experiments and were analyzed by one-way ANOVA and Student’s *t*-test. * *p* < 0.05, ** *p* < 0.01, **** *p* < 0.0001, NS = not significant.

**Figure 4 cells-10-03510-f004:**
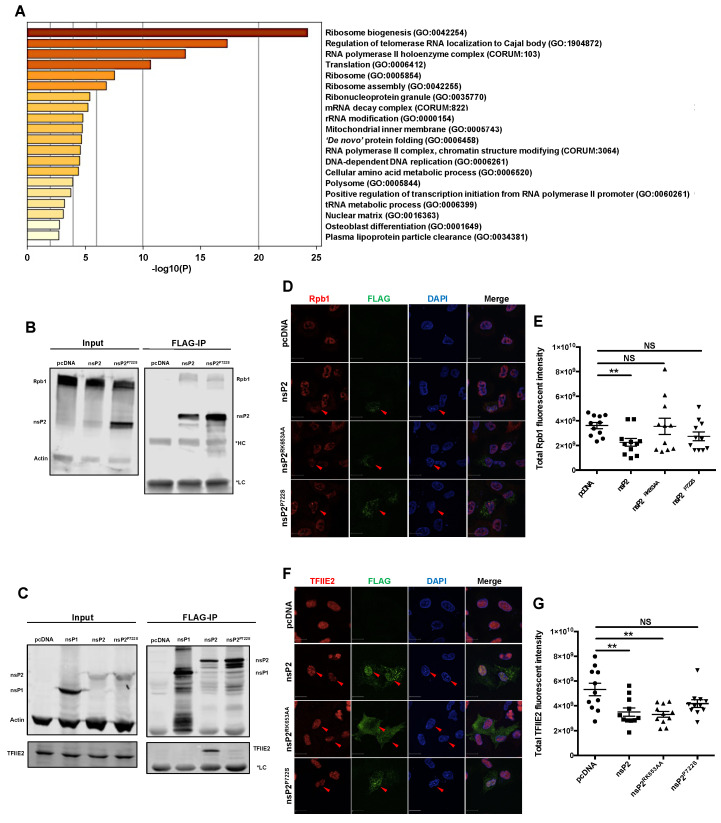
nsP2 interacted with key transcription mediators, Rpb1 and TFIIE2, and downregulated their expression. (**A**) Heatmap of GO term enrichment analysis. HEK293T cells were transfected with pcDNA empty vector or nsP2 for 24 h, after which whole cell lysates were immunoprecipitated using magnetic beads coated with anti-FLAG. nsP2-interacting proteins were identified by liquid chromatography–tandem mass spectrometry, and the protein–protein interactions from three independent experiments were scored by MiST analysis. nsP2-interacting host proteins with MiST score of ≥0.8 were selected to perform the GO enrichment analysis using the Metascape software. (**B**,**C**) HEK293T cells were transfected with the indicated viral protein constructs, and 24 h later, whole cell lysates were immunoprecipitated with magnetic beads coated with anti-FLAG and then subjected to immunoblot analysis using antibodies against FLAG, Rpb1, and actin (**B**), or FLAG, TFIIE2, and actin (**C**). Representative blots from three independent experiments are shown. (**D**,**E**) WT MAYV nsP2 downregulated Rpb1. A549 cells were transfected with nsP2 WT or nsP2^RK653AA^ (NLS-deficient), or nsP2^P722S^ (host transcription shutoff deficient) constructs. After 24 h, cells were fixed and stained using α-FLAG and α-Rpb1 antibodies and imaged using a confocal microscope using 60× oil objective. The total Rpb1 fluorescent intensities were quantified using Volocity software. Scale bar = 12 µm. Data represent the mean ± SEM from three independent experiments (*n* = 11) and were analyzed by one-way ANOVA and Student’s *t*-test. ** *p* < 0.01, NS = not significant. (**F**,**G**) A549 cells were transfected with indicated nsP2 constructs. After 24 h, cells were fixed, stained using antibodies to FLAG and TFIIE2, and visualized with a confocal microscope using 60× oil objective. The total TFIIE2 fluorescent intensities were quantified using Volocity software. Scale bar = 12 µm. Data represent the mean ± SEM from three independent experiments (*n* = 11) and were analyzed by one-way ANOVA and Student’s *t*-test. ** *p* < 0.01, NS = not significant.

**Figure 5 cells-10-03510-f005:**
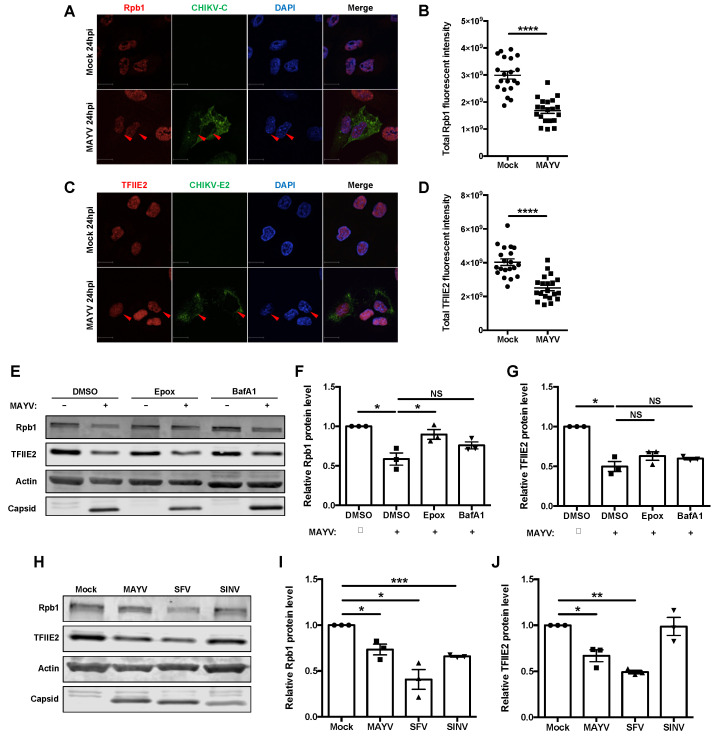
Reduction of RNA polymerase II complex factors during alphavirus infection. (**A**–**D**) A549 cells were infected with MAYV (MOI = 1) for 24 h and then fixed, stained with antibodies to CHIKV-capsid and Rpb1 (**A**,**B**) or TFIIE2 (**C**,**D**), and then images were acquired using confocal microscope equipped with 60× oil lens. The total Rpb1 and TFIIE2 fluorescent intensities were quantified using Volocity software. Scale bar = 12 µm. Data represent the mean ± SEM from three independent experiments (*n* = 20) and were analyzed by Student’s *t*-test. **** *p* < 0.0001. (**E**–**G**) A549 cells were infected with MAYV (MOI = 3) for 8 h, then treated with epoxomicin or bafilomycin (100 µM) for 24 h. Cell lysates were then subjected to immunoblotting using antibodies to Rpb1, TFIIE2, actin, CHIKV-capsid, and SFV-capsid proteins. The intensities of the protein bands were quantified using Image Studio software and then normalized to actin levels and expressed as folds of mock infected control. Data represent the mean ± SEM from three independent experiments and were analyzed by one-way ANOVA and Student’s *t*-test. * *p* < 0.05, NS = not significant. (**H**–**J**) A549 cells were infected with either MAYV, SFV, or SINV (MOI = 1) for 24 h, after which cell lysates were processed for immunoblotting using antibodies against Rpb1, TFIIE2, actin, CHIKV-capsid, and SFV-capsid proteins. The intensities of the protein bands were quantified using Image Studio software and then normalized to actin levels and expressed as folds of mock infected control. Data presented represent the mean ± SEM from three independent experiments and were analyzed by Student’s *t*-test. * *p* < 0.05, ** *p* < 0.01, *** *p* < 0.001.

**Figure 6 cells-10-03510-f006:**
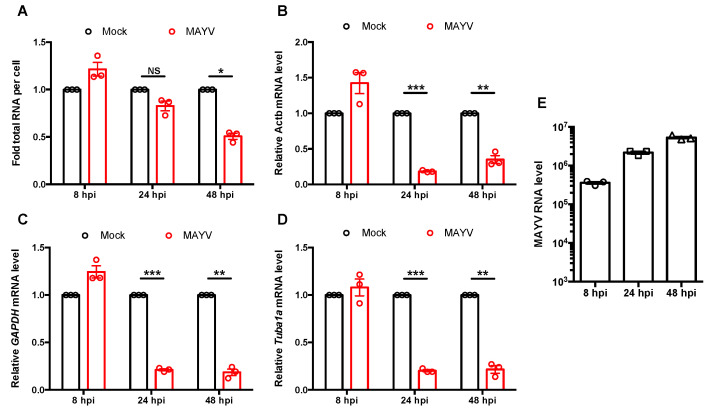
MAYV blocked global transcription during infection. (**A**–**E**) A549 cells were infected with MAYV (MOI = 3) for 8, 24, and 48 h, after which total cellular RNA was extracted and quantified using a spectrophotometer (**A**). Relative levels of *ACTB* (**B**), *GAPDH* (**C**), *Tuba1a* (**D**), and MAYV RNA (**E**) were measured by qRT-PCR, normalized by cell count, and expressed as fold mock infected cells. Data shown represent the mean ± SEM from three independent experiments and were analyzed by Student’s *\*-test. * *p* < 0.05, ** *p* < 0.01, *** *p* < 0.001, NS = not significant.

## Data Availability

All relevant data generated during this study are included in this article or Appendix A. Raw data and relevant information are available from the corresponding author upon request.

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
