# Peer review of "Mayaro Virus Non-Structural Protein 2 Circumvents the Induction of Interferon in Part by Depleting Host Transcription Initiation Factor IIE Subunit 2"

_cells, 2021, doi:10.3390/cells10123510_

Round 1

Reviewer 1 Report

The study by Ishida et al. investigated the mechanism of MAYV suppression of the interferon induction pathway in an in vitro system. The authors demonstrated that MAYV infection leads to a reduction of type I and type III IFNs at a transcript level, resulting in reduced secretion of IFN from infected cells. Further investigation showed that the non-structural protein 2 (nsP2) plays a role in shutting down IFN induction pathway downstream of IRF3 nuclear translocation and that the nsP2 helicase and protease domains are important in antagonising IFN induction.

nsP2 was found to translocate to the nucleus and form complexes with RNA polymerase II subunit A (Rbp1), thereby downregulating levels of Rpb1 in MAYV infection. The authors also discovered a novel interaction between nsP2 and host transcription mediator, transcriptional initiation factor IIE subunit 2 (TFIIE2), whereby this stable interaction depletes level of TFIIE2 in infection. Depletion of these transcription mediators by nsP2 has led to reduced total cellular RNA, increasing levels of viral RNA, hence concluding that IFN induction is overridden by blocking global transcription.

Overall, manuscript’s data are convincing and are backed up by previous findings on alphavirus studies. Conclusions made are appropriate based on presented results.

Manuscript is well written with appropriate use of language. While the results are solid, it was confusing to read and the section could use better structure. For example, in result section 3.3, lines 363 to 373 would read better if included at the end of 3.2 as this section mainly discussed nsP2 as a viral protein of importance. The change of focus from nsP2 domain to transcriptional shutoff seemed too abrupt in 3.3. In addition, line 374 to 389 would flow better as the first paragraph of section 3.4, as in 3.4 they discuss interaction of nsP2 and host transcription mediators. Section 3.4 also has a more solid and coherent title than 3.3. Otherwise, line 374 to 389 could also work as a standalone section. Subheading of 3.5 does not reflect the main content/message of the section.

The unmerged confocal microscopy result images on pages 7,9, 11, 13, and 15 should be brighter (more colourised?) otherwise it is hard to see staining/?

Line 368: Inconsistency in what is in text and in bracket. 

Author Response

We thank the reviewer for their comments and constructive suggestions. We have incorporated the suggestions and clarified some of the questions raised. We are confident that the changes made to the manuscript and the replies for the questions will address all the concerns raised by the reviewer. The responses to the reviewer comments are shown below.

Comments and Suggestions for Authors

The study by Ishida et al. investigated the mechanism of MAYV suppression of the interferon induction pathway in an in vitro system. The authors demonstrated that MAYV infection leads to a reduction of type I and type III IFNs at a transcript level, resulting in reduced secretion of IFN from infected cells. Further investigation showed that the non-structural protein 2 (nsP2) plays a role in shutting down IFN induction pathway downstream of IRF3 nuclear translocation and that the nsP2 helicase and protease domains are important in antagonising IFN induction.

nsP2 was found to translocate to the nucleus and form complexes with RNA polymerase II subunit A (Rbp1), thereby downregulating levels of Rpb1 in MAYV infection. The authors also discovered a novel interaction between nsP2 and host transcription mediator, transcriptional initiation factor IIE subunit 2 (TFIIE2), whereby this stable interaction depletes level of TFIIE2 in infection. Depletion of these transcription mediators by nsP2 has led to reduced total cellular RNA, increasing levels of viral RNA, hence concluding that IFN induction is overridden by blocking global transcription.

Overall, manuscript’s data are convincing and are backed up by previous findings on alphavirus studies. Conclusions made are appropriate based on presented results.

Manuscript is well written with appropriate use of language. While the results are solid, it was confusing to read and the section could use better structure.

For example, in result section 3.3, lines 363 to 373 would read better if included at the end of 3.2 as this section mainly discussed nsP2 as a viral protein of importance. The change of focus from nsP2 domain to transcriptional shutoff seemed too abrupt in 3.3.

Response: This section has been revised as suggested.

In addition, line 374 to 389 would flow better as the first paragraph of section 3.4, as in 3.4 they discuss interaction of nsP2 and host transcription mediators. Section 3.4 also has a more solid and coherent title than 3.3. Otherwise, line 374 to 389 could also work as a standalone section. Subheading of 3.5 does not reflect the main content/message of the section.

Response: This section has been revised as suggested.

The unmerged confocal microscopy result images on pages 7,9, 11, 13, and 15 should be brighter (more colourised?) otherwise it is hard to see staining/?

Response: We increased the brightness of the images as the reviewer suggested.

Line 368: Inconsistency in what is in text and in bracket. 

Response: This error has been corrected.

Reviewer 2 Report

Ray Ishida and his colleagues has revealed how Mayaro virus infection affects the induction phase of the IFN response and their findings suggested that nsP2-mediated inhibition of host cell transcription is an important aspect of how some alphaviruses block IFN induction. Overall, the experimental designs and the findings were reasonable and sound scientifically acceptable. Please see my comments below.

Lines 66 - 67: Statement is not clear. Based on the 3 references cited [28-30], only reference 28 stated the infection of MAYV on fibroblasts.

Lines 250 - 252: Even though both IFN-α and IFN-β belong to type I IFN, it might be good if IFN-β, instead of IFN-α, is used to treat the cells for the inhibition of MAYV.

Fig. 4: The bands of Actin on immunoblot (4B) were faint, but it seems working in Fig. 3F, 4C, 5E and 5H. Please clarify if it was due by competitive of multi-antibodies staining or other effects? It should be noted in materials and methods, and results.

According to your finding, the blocking of IFN induction by nsP2 to Rbp1 and/or TFIIE2 is a direct or indirect interaction?

Line 55: poprotein 5 -> protein 5

Lines 55, 64, 72 & 58, 121, 504, 532: signaling or signalling?

Line 73: punctuation mark, full stop

Line 76: indicate -> indicated

Line 81: complex, suggesting

Line 90: was -> were

Line 96: SINV: no full form (Sindbis virus) stated before the abbreviation

Line 116: anti-CHIKV-capsid & anti-SFV-Capsid

Lines 189 & 192: Blocking buffer & blocking buffer

Line 190: incubated in at room temperature

Line 205: primer sequences

Line 224: luciferase activity was measured

Line 317: nsp2 -> nsP2

Lines 387 & 471: suggest -> suggested

Author Response

We thank the reviewer for their comments and constructive suggestions. We have incorporated the suggestions and clarified some of the questions raised. We are confident that the changes made to the manuscript and the replies for the questions will address all the concerns raised by the reviewer. The responses to the reviewer comments are shown below.

Comments and Suggestions for Authors

Ray Ishida and his colleagues has revealed how Mayaro virus infection affects the induction phase of the IFN response and their findings suggested that nsP2-mediated inhibition of host cell transcription is an important aspect of how some alphaviruses block IFN induction. Overall, the experimental designs and the findings were reasonable and sound scientifically acceptable. Please see my comments below.

Lines 66 - 67: Statement is not clear. Based on the 3 references cited [28-30], only reference 28 stated the infection of MAYV on fibroblasts.

Response: We have rephrased the sentence to clarify the point that we are referring to multiple alphaviruses. ‘secretion of IFNs is almost completely abrogated during infection of fibroblasts by multiple alphaviruses [28-30]’

Lines 250 - 252: Even though both IFN-α and IFN-β belong to type I IFN, it might be good if IFN-β, instead of IFN-α, is used to treat the cells for the inhibition of MAYV.

Response: As both IFN-α and IFN-β signal through the same pathway, they are used interchangeably in literature to study type I IFN signaling. We have used IFN-α previously to study the effect of other pathogens such as Zika virus (PMID: 27797853) and SARS-CoV-2 (PMID: 34110264) on type I IFN signaling.

Fig. 4: The bands of Actin on immunoblot (4B) were faint, but it seems working in Fig. 3F, 4C, 5E and 5H. Please clarify if it was due by competitive of multi-antibodies staining or other effects? It should be noted in materials and methods, and results.

Response: Rpb1 and beta-actin were stained with corresponding anti-mouse antibodies on the same blot and were detected with anti-mouse secondary antibody. Due to the strong signal for Rpb1, we had to lower the acquisition sensitivity to keep the signal of Rpb1 in linear range, which in turn reduced the intensity of beta-actin signal.

According to your finding, the blocking of IFN induction by nsP2 to Rbp1 and/or TFIIE2 is a direct or indirect interaction?

Response: It is likely that the virus uses a global strategy of blocking transcription, which will inhibit any transcription-dependent cellular antiviral defense including IFN induction. We consider this an indirect strategy as it is not specifically targeting the IFN response alone.

Line 55: poprotein 5 -> protein 5

Response: Thank you for bringing this to our attention. We revised the text to indicate that MDA5 is melanoma differentiation associated protein 5.

Lines 55, 64, 72 & 58, 121, 504, 532: signaling or signalling?

Response: Revised text to state ‘signaling’

Line 73: punctuation mark, full stop

Response: This error has been corrected.

Line 76: indicate -> indicated

Response: This error has been corrected.

Line 81: complex, suggesting

Response: Suggested change was made to text.

Line 90: was -> were

Response: Suggested change was made to text.  

Line 96: SINV: no full form (Sindbis virus) stated before the abbreviation:

Response: We added the full virus name ‘Sindbis virus’ as suggested by the reviewer.

Line 116: anti-CHIKV-capsid & anti-SFV-Capsid

Response: We changed it as recommended by the reviewer to ‘anti-SFV-capsid’.  

Lines 189 & 192: Blocking buffer & blocking buffer

Response: We changed it as recommended by the reviewer to ‘blocking buffer’.

Line 190: incubated in at room temperature

Response: We changed it as suggested by the reviewer to ‘at’.

Line 205: primer sequences

Response: We changed it as recommended by the reviewer.   

Line 224: luciferase activity was measured

Response: We changed it as suggested by the reviewer.  

Line 317: nsp2 -> nsP2

Response: We changed it to ‘nsP2’ as suggested by the reviewer.  

Lines 387 & 471: suggest -> suggested

Response: We changed it to ‘suggested’ as advised by the reviewer.  

Reviewer 3 Report

This paper described the role of Mayaro virus nsP2 on the IFN response.  The experiments were clever and nailed down where in the IFN pathway MAYV acts in comparison with other alphaviruses.  Experiments on the role of nsP2 on TFIIE2 identified potential mechanisms for MAYV pathogenesis and its unique virology in comparison to other alphaviruses.

The methods were sound and all results were supported by multiple measurements and approaches.

I only have 2 comments.

Figures 1I, 2F, 3D, 4D, 4F, 5A, 5C: please make all lanes color.

3.1 and 3.2: please explain why you used sendai virus in your experiments.

Author Response

We thank the reviewer for their comments and constructive suggestions. We have incorporated the suggestions and clarified some of the questions raised. We are confident that the changes made to the manuscript and the replies for the questions will address all the concerns raised by the reviewer. The responses to the reviewer comments are shown below.

Comments and Suggestions for Authors

This paper described the role of Mayaro virus nsP2 on the IFN response.  The experiments were clever and nailed down where in the IFN pathway MAYV acts in comparison with other alphaviruses.  Experiments on the role of nsP2 on TFIIE2 identified potential mechanisms for MAYV pathogenesis and its unique virology in comparison to other alphaviruses.

The methods were sound and all results were supported by multiple measurements and approaches.

I only have 2 comments.

Figures 1I, 2F, 3D, 4D, 4F, 5A, 5C: please make all lanes color.

Response: We changed the lane colors as suggested by the reviewer.

3.1 and 3.2: please explain why you used sendai virus in your experiments.

Response: In our hands, Sendai virus induced a more robust IFN response with less variability, compared to poly(I:C). This was especially evident in our immunofluorescence assays where we observed a more uniform IRF3 nuclear localization in samples treated with Sendai virus compared to poly(I:C).  We have explained this in lines 259 and 263 of the text.